# Recent Strategic Advances in CFTR Drug Discovery: An Overview

**DOI:** 10.3390/ijms21072407

**Published:** 2020-03-31

**Authors:** Marco Rusnati, Pasqualina D’Ursi, Nicoletta Pedemonte, Chiara Urbinati, Robert C. Ford, Elena Cichero, Matteo Uggeri, Alessandro Orro, Paola Fossa

**Affiliations:** 1Department of Molecular and Translational Medicine, University of Brescia, 25123 Brescia, Italy; marco.rusnati@unibs.it (M.R.); chiara.urbinati@unibs.it (C.U.); 2Institute for Biomedical Technologies, National Research Council (ITB-CNR), 20090 Segrate (MI), Italy; pasqualina.dursi@itb.cnr.it (P.D.); matteo.uggeri@itb.cnr.it (M.U.); alessandro.orro@itb.cnr.it (A.O.); 3UOC Genetica Medica, IRCCS Istituto Giannina Gaslini, 16147 Genova, Italy; nicoletta.pedemonte@unige.it; 4Faculty of Biology, Medicine and Health, University of Manchester, Manchester M13PL, UK; robert.ford@manchester.ac.uk; 5Department of Pharmacy, Section of Medicinal Chemistry, School of Medical and Pharmaceutical Sciences, University of Genoa, 16132 Genoa, Italy; cichero@unige.it

**Keywords:** CFTR, drug discovery, bioinformatics, biosensors

## Abstract

Cystic fibrosis transmembrane conductance regulator (CFTR)-rescuing drugs have already transformed cystic fibrosis (CF) from a fatal disease to a treatable chronic condition. However, new-generation drugs able to bind CFTR with higher specificity/affinity and to exert stronger therapeutic benefits and fewer side effects are still awaited. Computational methods and biosensors have become indispensable tools in the process of drug discovery for many important human pathologies. Instead, they have been used only piecemeal in CF so far, calling for their appropriate integration with well-tried CF biochemical and cell-based models to speed up the discovery of new CFTR-rescuing drugs. This review will give an overview of the available structures and computational models of CFTR and of the biosensors, biochemical and cell-based assays already used in CF-oriented studies. It will also give the reader some insights about how to integrate these tools as to improve the efficiency of the drug discovery process targeted to CFTR.

## 1. Cystic Fibrosis: Pathogenesis, Clinical and Therapeutic Implications

During the past fifteen years the cystic fibrosis transmembrane conductance regulator (CFTR) has been the subject of a dramatic increase of published papers, capturing increasing attention as a therapeutic target. In fact, cystic fibrosis (CF) although classified as a rare disease, is the most common monogenic disorder in people with primarily European origins, leading to an estimate of 1% of the world’s population carrying one defective copy of the CFTR gene [1]. It is caused by different mutations that can occur in different domains of the protein. Among the 2000 genetic variants identified so far that can cause a reduced or loss-of-function phenotype, about 280 have been reported to be pathogenic. The disease-causing mutations have been classified into six different classes (class I–VI) based on the mechanism by which they disrupt CFTR production and function [2]. Among these, the deletion of phenylalanine at position 508 (F508del) is undoubtedly the most frequent one. F508del causes CFTR dysfunction by multiple mechanisms [3]. Indeed, F508 is located at the surface of the nucleotide-binding domain 1 (NBD1), at the interface with ICL4 and ICL1 [4]. F508del disrupts the interaction between NBD1 and ICL4 [5], which is important for the assembly of CFTR [5,6]. F508del also affects the folding kinetics of NBD1 by decreasing or delaying NBD1 folding efficiency [7] in molecular dynamic simulations [8]. This, in turn, might increase F508del-CFTR recognition capacity of the endoplasmic reticulum quality control machinery. In this way, F508del eventually hampers CFTR processing and intracellular trafficking to the plasma membrane [9,10]. Additionally, the mutation can destabilize CFTR protein when expressed at the plasma membrane, interfering with channel gating and reducing open channel probability [11,12].

CF has a major impact on the respiratory system, with patients that produce a thick mucus that cannot be correctly cleared and show impairment of innate defense mechanisms against bacteria [13]. The resulting pulmonary infections cause a progressive loss of respiratory function that, at the last stage of the disease, requires a lung transplant as life-saving intervention [14,15].

Current CF therapies fall in two categories: symptomatic treatments, consisting in the administration of aggressive antibiotics, osmotic agents and pancreatic enzyme products and systematic treatments, consisting in the administration of CFTR modulators able to improve the deficient or defective activity of the mutated CFTR by either restore its trafficking (correctors) or gating (potentiators) [16]. A third class of CFTR modulators is represented by amplifiers, whose mechanism of action is distinct from (and complementary to) those of correctors and potentiators since they stabilize and increase CFTR mRNA [17].

Systematic treatment of CF represents an almost mandatory future development on which both pharmaceutical companies and academic research groups are already making a big effort. In effect, even if symptomatic treatments have significantly pushed forward the mean survival age from 6 months (in 1938) to 47 years (in 2000), life quality and expectancy for CF patients still need to be ameliorated, by enriching the therapeutic options and the possibility of personalized therapies (mainly based on the type of CFTR mutation). Unfortunately, only four registered compounds are currently on the market: VX809, VX770, VX661, VX445 and their combinations [18], calling for the discovery of new CF drugs, possibly more efficacious.

Drug discovery is a time-consuming and expensive process. It is usually carried on by screening large compounds libraries through high throughput screening approaches or by rational design methods. These require appropriate experimental models to validate the therapeutic potential of the new putative drugs and then preliminary biological assays and eventually clinical trials to evaluate the safety and true efficacy of the drug. In the last years, many efforts have been made to rationalize, speed up and lower the costs of the drug discovery process. Relevant to this point, computational methods have become indispensable tools in the drug discovery process for many important human pathologies. In particular, the identification of drug–target interactions in silico has become an almost mandatory step in early stages of drug discovery [19]. However, once identified, the virtual drug–target interactions need to be experimentally validated and characterized, a task for which the use of biosensors has demonstrated to be particularly suitable [20].

Bioinformatics and biosensors have been already used in the study of CFTR, pointing to the necessity of an integrated approach aimed at the devising of a fast, reliable pipeline of analysis to speed up the identification of new CFTR-targeted drugs. This, in turn, requires the availability of appropriate computational models of the mutated CFTR structure and the development of dedicated biosensor and biochemical assays to be flanked to the already available cell-based assays.

Aimed at giving an updated picture of the field, this review will report and discuss the recent insights gained on the 3D structure of the full-length CFTR protein, the most recently developed computational models as well as the most important biochemical and biological assays and their appropriate integration to speed up the discovery of new drugs for the treatment of CF.

## 2. CFTR Experimental Structures

CFTR is a large membrane-integral protein, belonging to the ATP-binding cassette superfamily (ABC) and is composed of two intracellular NBDs, one regulatory domain (R), and two transmembrane domains (MSDs) that span the cell membrane six times each [21] (Figure 1). Despite this complexity, the first and still most abundant structural models for CFTR refer to NBD domains, especially to NBD1, with only a structure for human wild type (WT) NBD2 complexed with N6-phenylethyl ATP reported in the protein data bank (PDB) (code 3GD7). In effect, the PDB (protein data bank) lists 33 structures of the CFTR NBD1, among which 24 are human and 9 murine (Table 1).

To complete the structural data on CFTR, four complexes between 14-3-3 proteins and some phosphorylated motifs of the intrinsically disordered R-domain are available (PDB codes 5D2D, 5D3E, 5D3F and 6HEP). The binding of the 14-3-3 proteins to the R domain is able to increase the trafficking of CFTR to the membrane. The interaction between the 14-3-3 proteins and R phosphorylated motifs has been reported in details [30]. These models, together with the homology models of the whole protein (see Section 3), have greatly helped in elucidating the 3D structure of the whole protein and have triggered the pioneer drug design studies on CFTR.

In the last four years the complete structure of CFTR has been available, due to cryoelectron microscopy (cryo-EM) [31]. This technique has broadened horizons for structural studies in molecular and cell biology, pushing forward the structure determination of subcellular structures. In particular, the research in the field of CFTR has took advantage of the structural information provided by cryo-EM, since the eight WT structures available in the Protein Data Bank and listed in Table 2, have all been determined with this technique.

In conclusion, to date a number of 3D structures for CFTR are available to investigate ligand–target interactions at a molecular level and to perform structure-based drug design studies. However, there is still a lot of work to be done to fix some important aspects:(i)Some of the NBD1 structures available (Table 1) contain stabilizing or solubilizing point mutations introduced to get the crystals.(ii)All the reported structures of intact CFTR (Table 2) refer to the WT protein and not to disease variants, thus information concerning the different flexibility in the 3D structure of the mutated pathological proteins (e.g., F508del-NBD1 or -CFTR) are scarcely available for drug discovery. Additionally, the chicken structures show several mutations, deriving from the thermostabilization process of the protein, necessary for performing the structural studies [35];(iii)The dephosphorylated structures, ATP-free or ATP-bound refer to an inactive state of the protein, being phosphorylation a fundamental step in CFTR activation;(iv)All structures miss the secondary structure assignation of large parts of the protein (Phe409-Gly437, Gln637-Trp845 and Gly1173-Asp1202) and in some cases the R-domain, which contains multiple phosphorylation sites, essentials for regulating the channel activation after phosphorylation;(v)Different outward-occluded or inward-facing protein conformations can be observed under very similar experimental conditions [38]. As a consequence, the flexibility of the whole protein, as well as that of its subdomains, deserve further studies, being it an aspect of great importance in understanding the CFTR–ligand molecular mechanism of interaction. Figure 2 shows the X-Ray structures of CFTR colored by B-factor values, which indicate the static or dynamic mobility of an atom, with higher values corresponding to large fluctuations.

## 3. CFTR Computational Models

Due to the lack of a complete 3D structure for F508del-CFTR and the need of a representative model of the protein in its active state, a number of computational models have been developed based on CFTR structural similarity with other ABC transporters.

The first brick to develop these 3D models has been the crystal structure of the full ABC, multidrug transporter Sav1866 from *Staphylococcus aureus* [39]. The protein, in its outward-facing conformation, has been used as a template for modeling the outward configuration of CFTR.

The first model to be developed on the basis of this experimental structure [5] was a mix of computational and experimental data assembled by inserting the missing regulatory loop computationally built on the existing crystal structures of NBD1 (PDB code 2BBO), while the NBD1-NBD2 dimer was constructed from NBD1 crystal structure and the homology model of NBD2 (obtained using as template ABC transporters) superimposed on the Sav1866 structure (PDB code 2HYD). The whole model refers to WT CFTR protein in its active (closed) state [5]. In the same year the model by Mornon [40] was constructed aligning the human CFTR sequence (MSDs and NBDs) with the *S. aureus* Sav1866 sequence. The resulting 3D structure is a pure Sav1866-based model, that, despite taking into account the high structural plasticity of the NBDs-MSDs interfaces, refers to WT CFTR and misses the R-domain, thus representing an improved, but still a limited model.

A third CFTR model with the protein in its inward-facing conformation was again proposed by Mornon [4] based on several structures of the ABC lipid flippase MsbA and with the modeling of the R-domain. This closed-CFTR model has represented the first complete structure for investigating the molecular basis of CFTR functioning. However, it referred to a WT protein, so that no information on structural modifications determined by mutations (e.g., F508del) may be derived. Additionally, the model was not validated by molecular dynamics studies (MDs), leaving out information on protein flexibility, very important especially after F508 deletion.

Following these pioneering computational studies, various authors [41,42,43] investigated with MDs the dynamic aspects of CFTR opening and closing by means of models built using the template structure of Sav1866 or an “hybrid” template obtained by joining together available experimental structures for the NBDs and MSDs. Among these contributions the work by Furukawa-Hagiya has the merit to propose the first F508del-CFTR 3D model [43].

The efforts to derive a predictive model of the most common mutation F508del-CFTR and to investigate its dynamic behavior in comparison with the WT protein were pursued by Mornon [44], and Belmonte [45]. In addition, Corradi [46] elaborated four WT CFTR homology models based on the templates TM (287–288), ABC-B10, McjD and Sav1866, so as to explore possible conformational states visited by CFTR during its gating and to get more information on the structural properties of the transmembrane cavity. A further contribution in unraveling CFTR structure is represented by the first study, which used the available cryo-EM map of the human protein derived from 2D crystals for refining two previously reported homology models of CFTR [47].

Among the computational models listed above, only a few have been used to better characterize the binding mode of known F508del-CFTR-rescuing drugs, to virtually screen libraries of molecules aimed at the identification of new putative F508del-CFTR-binding compounds or to design new CFTR-targeted drugs [48,49]. On the contrary, a huge amount of computational studies has been developed in the attempt of clarifying at a molecular level the mechanism of action of F508del-CFTR-ligands, using several NBDs RX data [23,50,51,52] or the more recent cryo EM structures [53].

## 4. Biochemical and Biological Assays to Screen for CFTR Modulators

As mentioned above, appropriate computational models of F508del-CFTR have been already used to screen virtual libraries of molecules to identify new F508del-CFTR-binding/rescuing compounds. In silico predicted hits need however an experimental validation that can be provided by appropriate biochemical or biological (cell-based) assays that will be described here below.

### 4.1. Binding Assays: Surface Plasmon Resonance (SPR) Spectroscopy

Surface plasmon resonance (SPR) is a solid-phase optical-based technology that allows the evaluation of the interaction of biological macromolecules with putative drugs (Figure 3).

Briefly, in the SPR technology, a polarized beam of visible monochromatic light is passed through a prism fitted with a gold film attached to a glass. When the light hits the glass, an electric field (evanescent wave) is generated and absorbed by free electrons in the gold film, reducing the intensity of the light that, once reflected by the gold is detected at the specular angle. The angle corresponding to the sharp intensity minimum, called resonance angle, depends on the refractive index of the material present within 300 nm from the gold surface.

In order to evaluate the interaction of two putative binders, one is immobilized onto the gold film (ligand) and then exposed to the second one injected in the fluidic system (analyte). The ligand/analyte interaction causes a variation of the refractive index at the gold surface and consequently of the resonance angle that is presented as a sensorgram, a real-time graph of the response units against time (Figure 3). For an exhaustive description of SPR technology see [54].

In comparison to conventional fluorescent, enzyme or radio-labeled binding assays, SPR is label-free and allows the manipulation of small amounts of molecules covering a wide range of molecular weights and binding affinities. Furthermore, SPR allows the evaluation of how fast the analyte binds to and detaches from its ligand and of the dissociation constant (K_d_) that is inversely proportional to the binding affinity. SPR is particularly suitable for the analysis of low molecular weight, weakly interacting drugs. Due to all these advantages, SPR has been successfully exploited for the study of a variety of pathologies and related drug discovery programs [55,56,57].

CF is one of the diseases for which SPR has been more frequently employed, surpassed only by hemophilia among hereditary diseases and by malaria and influenza among infectious diseases (Figure 4A). SPR analyses in CF have increased over the last twenty years (Figure 4B). As shown in Figure 4C, SPR has been mainly exploited for CF diagnosis [58] and for binding studies of CFTR with intracellular partners (Table 3) or putative drugs (Table 4).

Among natural binders, the most representative interactions studied by SPR have been: the C-terminal portion of CFTR with the multidomain scaffolding PDZ-based adaptor EBP50/NHERF1, that modulates the assembly and trafficking of CFTR [59,60]. The last eight amino acids of CFTR with different PDZ domains of CAP70, a protein involved in mediating inter-molecular CFTR interaction [61]. CFTR with the μ subunit of the endocytic clathrin adaptor AP-2, required for CFTR entering the endosomal compartment [62] and with the adaptor S100A10, that connects CFTR to annexin I and cytosolic phospholipase A2, forming a complex that modulates tumor necrosis factor-α-induced eicosanoids production [63]. CFTR with nucleoside diphosphate kinase B (NDPK-B; that controls chloride currents in epithelia [64]) and with calumenin (a chaperone involved in the trafficking pathway of the G551D-CFTR) [65]. Cytokeratin 8 (K8), that binds F508del-NBD1 with a higher affinity that the WT, suggesting that K8 contributes to F508del-CFTR retention and pointing to destruction of K8/F508del-CFTR interaction as a therapeutic target [63]. K8 fragments have been also used in SPR to map the binding interface of the K8/CFTR complex to design of specific inhibitors [68]. Similarly, also heat shock cognate 70 (HSC70) binds F508del-CFTR with higher affinity in respect to WT CFTR [14]. In this same work, SPR was used to evaluate the effect of ATP and ADP on the HSC70/CFTR binding.

SPR has been used for both the discovery of new CFTR-rescuing compounds and for the characterization of the binding modes of well-known CF drugs (Table 4):(i)In search for new CFTR-correctors/activators, SPR was used to evaluate the binding of CFTR to crotoxin from *Crotalus durissus terrificus* and the *Viperidae* snake venom PLA2, known to possess a large spectrum of pharmacological functions [72].(ii)SPR has been used to the study of the direct interaction with mutated CFTR of correctors VRT-325 and Corr-4a [14] and of either F508del-NBD1 [52] or intact F508del-CFTR [48] with a panel of AAT derivatives. Additionally, the pyrazole compound 4172 was demonstrated to bind F508del-CFTR with an affinity that is 25 times higher than that of the first identified type III corrector BIA [73].(iii)Some monoclonal antibodies have been analyzed by SPR for their interaction with a different domain of CFTR [14,70]. In particular, Gakhal et al. exploited SPR to characterize synthetic antigen-binding fragments (FABs) isolated from phage-displayed library specifically directed against different domains of CFTR [74].(iv)As already mentioned above, the binding of mutated CFTR to chaperones responsible for its retention is considered a therapeutic target to rescue CFTR activity. Relevantly, when tested in SPR competition assays, Corr-4a, VRT-325 and CFTRinh-172 effectively inhibit the binding of HSC70 to sensorchip immobilized F508del-CFTR [14,68].

As apparent by Figure 4D and Table 3 and Table 4, a large part of SPR studies have been so far directed only to NBD1, which may not contain the features needed to build high-affinity interactions [23], greatly limiting the identification of high-affinity CFTR-rescuing drugs. Relevant to this point, a purified form of human intact F508del-CFTR [75] has been recently used to identify new putative CF drugs and to characterize the CFTR-binding modes and/or molecular mechanisms of action of known CF drugs such as VX660, VX770 and VX809 [48,76].

In conclusion, SPR studies with CFTR have so far allowed the identification of a significant number of natural and synthetic binding partners. Moreover, they have made possible to draw a (still partial) “CFTR interactome” whose decoding may help to identify novel therapeutic targets and to design novel drugs for the cure of CF (Figure 5).

### 4.2. Biochemical Functional Assays: Thermostability Assay

Thermostability assays based on the Cys-reactive thiol-specific fluorochrome N-[4-(7-diethylamino-4-methyl-3-coumarinyl)phenyl]maleimide (CPM) have been introduced and reviewed. CPM is normally non-fluorescent or has low fluorescence in aqueous systems, but becomes intensely fluorescent upon formation of a covalent linkage with exposed Cys residues on the protein surface. Upon protein unfolding and denaturation, exposure of previously buried Cys residues allows further CPM binding with an increase of fluorescence that is a useful indicator of the stability of the protein. The initial labeling with CPM (at low temperature used for CFTR purification) gives an idea of the initial folded status of the protein and this initial labeling value can be greatly increased by the addition of chaotropes such as guanidium HCl and urea [38,76,77]. The subsequent denaturation of the protein can be followed by monitoring the increase in CPM fluorescence as more buried Cys residues become exposed, allowing the intrinsic thermostability of the protein to be assessed [76]. However, the CPM fluorescence in a system containing F508del CFTR at a physiological temperature can also provide information about the long term stability of the protein at 37 °C, a temperature where F508del CFTR is known to lose function rapidly [76]. No single assay for the screening of drugs and their effects on the biophysical properties and activity of a given protein will be perfect, and hence it is important to assess the strengths and weaknesses of thermostability assays that we have described in this manuscript. The use of tryptophan fluorescence for measuring the thermal stability of membrane proteins like CFTR has been described [78], but the levels of protein required to achieve a sufficient signal to noise to allow the determination of thermal unfolding events is high. The use of a Cys-reactive probe such as CPM allows a more sensitive assay to be employed that can distinguish relatively small shifts in thermal stability. Moreover, the high fluorescence output of the probe allows the minimization of the amount of protein required for each drug to be screened. For example, in the 48 well capillary format used in the UNCLE fluorimeter system (Unchained labs Inc.), a few tens of nanograms of protein are sufficient to achieve good signal to noise. However, such assays that involve the external addition of fluorescent reporter molecules, even when such a molecule is very small, inevitably lead to an increased likelihood of misleading results (false positives and negatives). Detergent is a major factor as CPM fluoresces once embedded in the hydrophobic portion of the micelle [79]. In the case of CPM, for both the isothermal and thermal unfolding assays, the other main type of misleading result occurs with drugs or natural products that also contain a sulphur atom that is capable of forming a covalent bond with CPM. It seems likely that false negative compounds form a covalent bond with CPM with fast kinetics, hence the initial labeling and fluorescence increase at low temperature is very rapid, and then following further heating some thermal quenching will be observed. In contrast false positives may appear to label quite slowly and show a slow fluorescence increase at lower temperatures, followed by a large increase in fluorescence as the kinetics of the reaction will be greater. In both these cases, the false hits can be identified in the raw fluorescence data (e.g., Figure 6) since these compound-CPM adducts will have significantly higher fluorescence at the 10 μM concentrations present than for the CFTR-CPM adducts, which will be in the 50–100 nM range.

For the CPM-based thermal unfolding assay, sulphur-containing drugs and natural products may give rise to false positives and false negatives as above. However, many plate reader fluorescence instruments use UV laser excitation source and this may also give rise to other types of false hits. For example, strongly fluorescent compounds or drugs and natural products with large delocalized electron systems may appear as false hits. In some cases, this may be because these compounds can lead to absorption of the UV light and anomalous behavior. This defect in the assay also means that ATP can only be employed at relatively low concentrations (e.g., at or below 1 mM) because nucleotides also strongly absorb UV light. Some false hits also appear to arise due to the intrinsic fluorescence of the drug or natural product, which can compete with the CPM signal. Ideally therefore, one should use a dedicated plate reader instrument with an excitation source closer to the CPM excitation maximum—around 384 nm.

### 4.3. Biological Assays to Screen for CFTR Modulators

Until a few years ago, considering the absence of solid structural data on CFTR and hence the location of druggable binding sites to rationally design F508del-CFTR-rescuing drugs, the most promising approach was to screen small molecule libraries searching for CFTR-rescuing compounds. To this aim, different functional or biochemical assays have been developed such as the direct detection of CFTR protein at the plasma membrane [80], electrophysiological techniques (including transepithelial electrical conductance and short-circuit current measurements [81]) and the evaluation of iodide efflux by means of ^125^I radioisotope [82], iodide-selective electrodes or chloride/iodide-sensitive fluorescent probes [83] (Figure 7).

Among anion-sensitive fluorescent probes, the microfluorimetric, functional assay based on a halide-sensitive yellow fluorescent protein (HS-YFP) [84] has shown a high sensitivity to several halides, in particular iodide, but reduced sensitivity to chloride [85]. This genetic probe allows the set-up of a convenient and robust transport assay that measures CFTR-mediated iodide transport into the cell as a function of HS-YFP fluorescence decrease over time. With this assay was identified the first CFTR corrector, namely corr-4a [86]. From there on, the assay has been extensively used to identify CFTR modulators including activators [87], inhibitors [88], potentiators [89,90,91,92,93,94], correctors [86,93,94,95], as well as proteostasis regulators (i.e., correctors that rescue F508del-CFTR trafficking defect by modulating proteins involved in CFTR maturation/degradation, thus leading to increased CFTR processing) [96].

In the last years, several modifications have been proposed to optimize the HS-YFP-based assay. Vijftigschild and co-workers [97] developed a dual sensor, radiometric assay, based on the coexpression of HS-YFP and an iodide insensitive fluorescent protein (the red fluorescent protein mKate) for normalization of sensor expression levels and accurate quantification of CFTR function. Although the usefulness of this assay for CFTR modulators screening has yet to be demonstrated, one of its sure advantages is that it could be used to measure CFTR function by flow cytometry in non-adherent cells [97], with an implication for the study of CFTR function in monocytes.

More recently, Langron and collaborators [98] developed two novel probes: YFP-F508del-CFTR (in which halide sensitive YFP is tagged to the N-terminal of CFTR), which provides a sensitive readout of the CFTR function, and CFTR-pHTomato (having a pH sensor tagged to the fourth extracellular loop of CFTR) to quantify the membrane and intracellular CFTR expression, independently of the function. YFP-F508del-CFTR has been used in a pilot screen of structural analogues of VX-770, while the F508del-CFTR-pHTomato assay was employed to assess long-term F508del-CFTR destabilization effects following chronic treatment with the hits arose from the first screening [98].

The YFP assay can therefore be exploited not only to screen libraries of compounds but also to characterize CFTR activity upon pharmacological treatment and to study mechanisms of action of compounds acting as CFTR modulators. Notwithstanding the improvement proposed by various researchers, its main weakness remains related to the use of an immortalized model for the primary screen, which is a common feature of the drug discovery projects run in the CF field. Indeed, it has been demonstrated that pharmacological correction of F508del-CFTR folding and trafficking defect is influenced by the cell background, with correctors that can be cell-type specific [99,100]. For this reason, the subsequent use of electrophysiological techniques to measure CFTR-mediated ion transport is required to further investigate novel putative CFTR modulators.

In conclusion, given the low throughput of electrophysiological assays, the availability of computational and biochemical methods able to efficiently make an initial selection and reduce the number of compounds to be tested with functional assays would (Figure 8) be a great asset.

## 5. Conclusions and Perspectives

To date, CF drug discovery is still largely based on drug screening by using well tried cell-based assays (Section 4.3). However, these assays are expensive, time-consuming and do not allow the characterization of the mechanisms of action of the drugs at a molecular level. To this aim, biochemical models such as SPR (Section 4.1) or the thermostability assay (Section 4.2) can be of help.

Of equal importance could be the availability of more complete structures of WT or mutated CFTR (Section 2) and the consequent development of reliable CFTR computational models that have been already used to investigate the different dynamic behavior of WT and mutated protein (Section 3) but have not been yet used at their full potential for CFTR-targeted drug discovery.

Finally, larger libraries of chemical fragments, natural products and already marketed drugs are now available (i.e., ZincDatabase, ChEMBL, ProteinDataBank, AIFA, FDA, Drugbank (which includes FDA, Health Canada and EMA drugs) and CHEMnetBASE-Dictionary of Natural Products) are available that, if properly exploited, would maximize the chances of identifying new CFTR-rescuing drugs.

However, given the high number of compounds present in these libraries, their screening cannot be done with the expensive and time-consuming biochemical or cell-based assays mentioned above. Instead, this enormous task becomes feasible by means of structural bioinformatics virtual screening, as already done for many other human diseases [101,102]. Indeed, only high performance computing (HPC) allows the handling of the huge amounts of data deriving from exhaustive conformational space sampling or from molecular dynamics simulations while investigating CFTR flexibility. Additionally, with HPC it is possible to deal with the complex algorithms and large volumes of data generated from large libraries screening.

Thus, the challenge of setting up an efficient, up-to-date CF drug discovery pipeline relays on an appropriate integration of HPC and the available experimental technologies aimed at decreasing the number of compounds under study with the increasing of the complexity and costs of the assays employed (Figure 8).

With this aim, the first phase of the drug discovery process must inevitably correspond to high-throughput computational screening, as to guarantee the capability to deal with even tens of thousands of compounds. Then, an intermediate filter to validate experimentally the computational prediction may be represented by biochemical (cell-free) systems such as SPR, that is fully automatable to a high-throughput level [103] and that has been already successfully used in CF-oriented studies (see Section 4.1). However, SPR can only provide a measure of the binding capacity of a compound that is not necessarily correlated to its CFTR-rescuing capacity. Once reduced the number of hits to a manageable level, the last phase of the CF drug discovery process must be performed with the cell-based assays described in Section 4.3 and schematized in Figure 7.

Important to note, the validity of the proposed pipeline can be already assessed retrospectively, by evaluating the degree of consensus of the prediction and results obtained with the various techniques employed. In effect, already reported cases of this consensus are: computational predictions vs. SPR [52] or vs. cell-based assays [49], immunoprecipitation vs. overlay assays or vs. SPR [70], SPR vs. thermostability assays or vs. cell-based assays [48].

A first practical application of this conceived pipeline for CF drug discovery is surely in drug repositioning technology. It is a method aimed at finding new applications for already marketed drugs or natural compounds. It reduces both the cost and duration of the drug development process and the likelihood of unforeseen adverse events. The most striking case of drug repositioning is thalidomide, a drug sold in the 1950s as a sedative that was removed from the market for causing severe birth defects and that later has been included in the treatments for multiple myeloma [104]. Concerning CF, drug repurposing has been applied only to ameliorate pulmonary delivery of inhaled antibiotics [105] and to enhance mucociliary clearance [106], but its exploitation to identify drugs able to correct misfolded CFTR has been to date only suggested [107,108], thus representing an open field of research.

As mentioned in the introduction, amplifiers are a class of CFTR modulators that cotranslationally increase CFTR mRNA stability by binding to poly(rC)-binding protein 1 [17], thus acting with a mechanism of action that is distinct from that of correctors and promoters that bind instead to CFTR itself. On this basis, an orthogonal cotherapy made up of amplifiers and correctors to be identified with the above described pipeline may enhance the efficacy of current CF therapies [109].

## Figures and Tables

**Figure 1 ijms-21-02407-f001:**
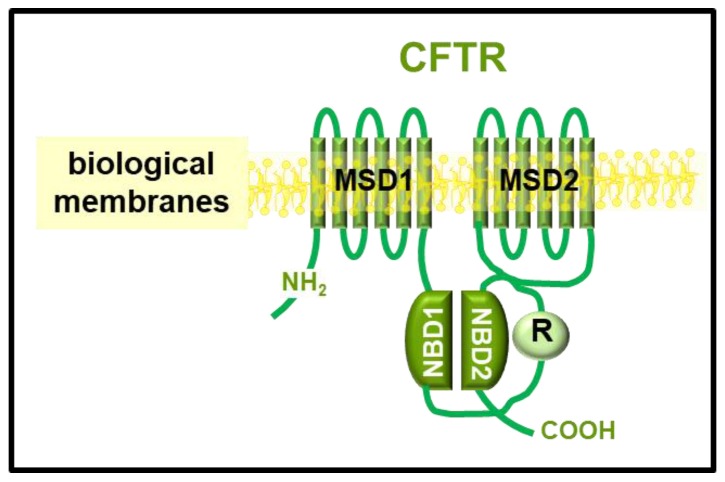
Schematic representation of the cystic fibrosis transmembrane conductance regulator (CFTR) structure. The different domains of the protein are indicated.

**Figure 2 ijms-21-02407-f002:**
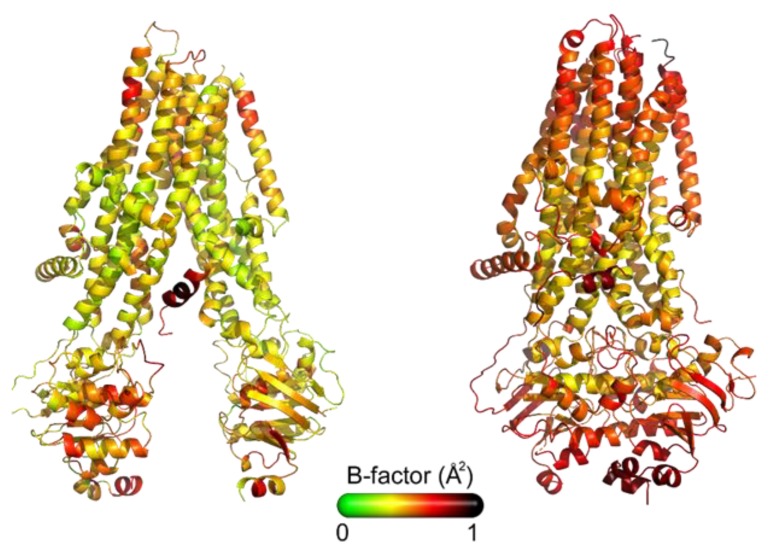
Ribbon diagram of CFTR X ray structures colored by B-factor values. Left: dephosphorylated CFTR wild type without ATP bound (PDB code 5UAK), right: phosphorylated CFTR (1 stabilizing mutation) with ATP bound (PDB code 6MSM). The color scale date range is shown below, the high and low values of CFTR mobility are in red and green respectively.

**Figure 3 ijms-21-02407-f003:**
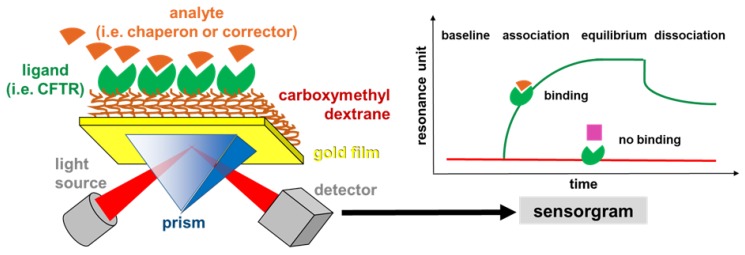
Schematic representation of surface plasmon resonance (SPR) technology. One of the two binders is immobilized onto the sensorchip (ligand), while the other (analyte) is injected into the microfluidic system. When occurring, the analyte/ligand interaction changes the refractive index of the gold surface to which the ligand is immobilized, providing label-free transduction of the binding event that is presented as a sensorgram, consisting in a real-time graph of the response units against time. In a sensorgram, three phases can be identified: association, equilibrium binding and dissociation.

**Figure 4 ijms-21-02407-f004:**
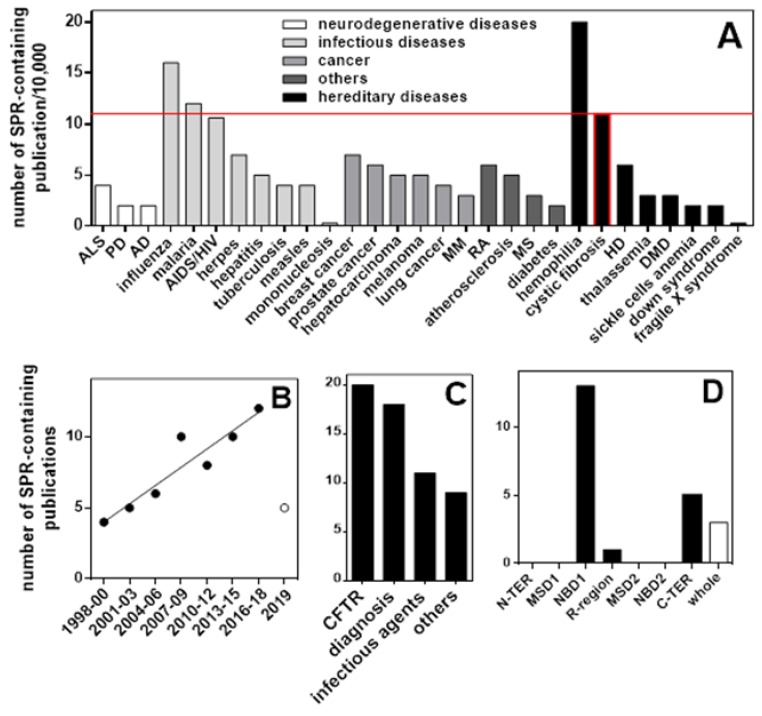
Contribution of SPR to cystic fibrosis (CF). (**A**) Use of SPR in various diseases. ALM: amyotrophic lateral sclerosis; PD: Parkinson disease; AD Alzheimer’s disease; MM: multiple myeloma; RA: rheumatoid arthritis; MS: multiple sclerosis; HD: Huntington’s disease; DMD: Duchenne muscular dystrophy. (**B**) Use of SPR in CF over time since its first reporting. Distribution of SPR analyses among the various topics in CF (**C**) or among the different modules of CFTR (**D**).

**Figure 5 ijms-21-02407-f005:**
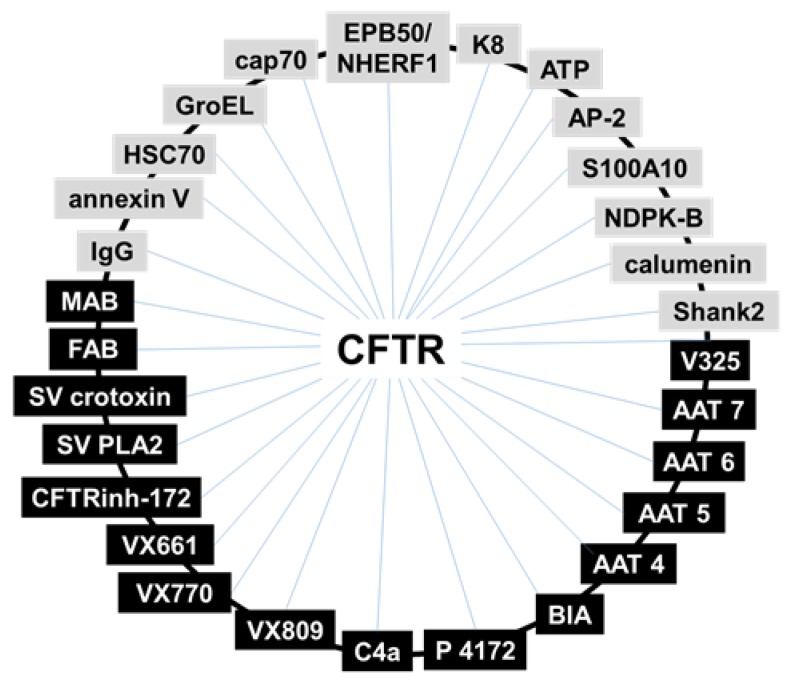
Schematic representation of CFTR intracellular interactome based on SPR studies. Natural intracellular CFTR-binders are indicated in grey, putative CFTR-rescuing drugs in black. P: pyrazole; SV: snake venom. See text for further details.

**Figure 6 ijms-21-02407-f006:**
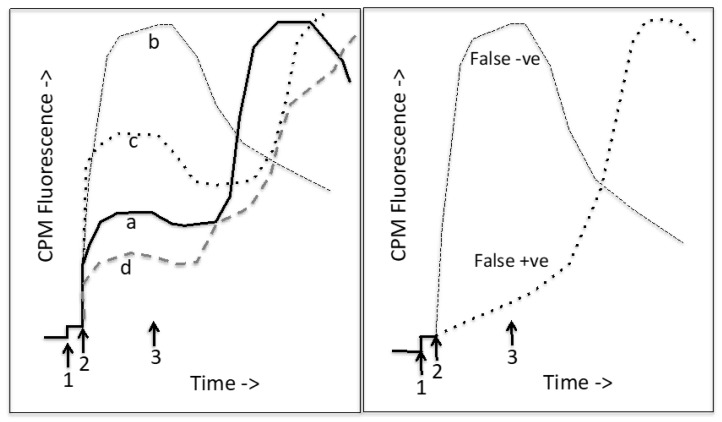
Thermostability assay. CPM labeling experiments: CPM is added to the cuvette (1) at low temperature (e.g., 10 °C) and then the protein is added (2). After a period of labeling of surface exposed Cys residues, the cuvette is heated (starting at 3). Initially the CPM fluorescence decreases due to thermal quenching, but then as the protein unfolds, the CPM fluorescence rises, reaching a maximum after the protein is completely unfolded. After this, thermal quenching of the CPM fluorescence is again observed. Left panel shows typical CPM unfolding profiles for different membrane proteins. Trace a (the solid line) is for a membrane protein such as CFTR with about 40% of its 18 Cys residues exposed to the aqueous medium. It shows a single cooperative unfolding transition. Trace b (black, dashed) is for an unusual (perhaps very small) membrane protein with no buried Cys residues. Trace c (black, dotted) is for a very thermophilic membrane protein but with mostly exposed Cys residues. Trace d is for a membrane protein with few exposed Cys residues in the native state. It has multiple domains that show separate unfolding transitions. The right panel illustrates potential false results when examining the effects of drugs. A false negative drug (false destabilizer) will have a CPM-reactive group that reacts rapidly with CPM at low temperature. A false positive drug (apparent corrector) will have a CPM-reactive group that labels slowly at low temperature and then the rate of reaction speeds up upon heating.

**Figure 7 ijms-21-02407-f007:**
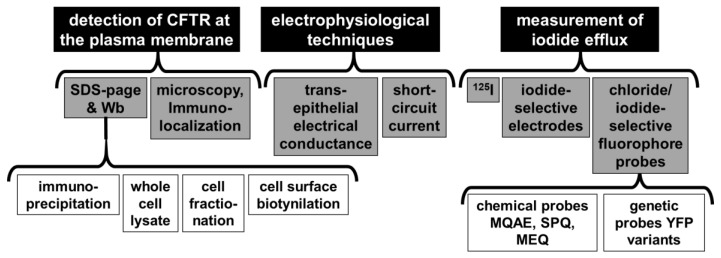
CFTR functional assays. The main cell-based assays to measure CFTR activity and F508del-CFTR rescue by drugs are reported. MQAE: (*N*-(ethoxycarbonylmethyl)-6-methoxyquinolinium bromide); SPQ: 6-methoxy-*N*-(sulfopropyl)quinolinium. MEQ: 6-methoxy-*N*-ethylquinolinium iodide.

**Figure 8 ijms-21-02407-f008:**
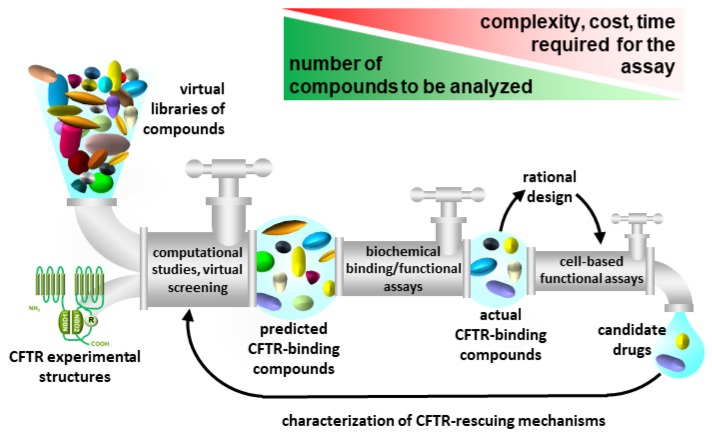
Integration of computational, biochemical and cell-based models in the pipeline for CF drug discovery. In the pipeline proposed, the number of compounds to be analyzed progressively decreases with the increasing of the complexity and costs of the analyses employed. The study of the chemical features of the hits may help the rational modification or de novo design of candidate drugs with improved binding affinity/rescuing capability. On the other hand, in deep computational docking and/or MDs of hits may help the comprehension of their CFTR-rescuing mechanisms.

**Table 1 ijms-21-02407-t001:** Structural data for human and mouse nucleotide-binding domain 1 (NBD1).

PDB Code	Organism	Resol. (Å)	Features	Complex with	Ref.
6GJS	*H. sapiens*	1.95	WT	ATP and 2 nanobodies	[22]
6GJU	*H. sapiens*	2.6	WT	nanobody	
6GJQ	*H. sapiens*	2.49	WT	nanobody	
6GK4	*H. sapiens*	2.91	WT	ATP and 2 nanobodies	
6GKD	*H. sapiens*	2.99	WT	ATP and 2 nanobodies	
4WZ6	*H. sapiens*	2.05	F508del, 3 stabilizing mutations	ATP	[23]
2PZE	*H. sapiens*	1.7	WT dimer	ATP	[24]
2PZF	*H. sapiens*	2.0	WT/F508del dimer	ATP	
2PZG	*H. sapiens*	1.8	WT	ATP	
2BBO	*H. sapiens*	2.55	F508del	ATP	[25]
2BBS	*H. sapiens*	2.05	WT, 3 stabilizing mutations	ATP	
2BBT	*H. sapiens*	2.3	WT, 2 stabilizing mutations	ATP	
1XMI	*H. sapiens*	2.25	WT	ATP	[26]
1XMJ	*H. sapiens*	2.3	F508del	ATP	
5TF7	*H. sapiens*	1.93	WT	ATP	[27]
5TF8	*H. sapiens*	1.86	WT	dTTP	
5TFA	*H. sapiens*	1.87	WT	dUTP	
5TFB	*H. sapiens*	1.87	WT	7-methyl-GTP	
5TFC	*H. sapiens*	1.92	WT	GTP	
5TFD	*H. sapiens*	1.89	WT	CTP	
5TFF	*H. sapiens*	1.89	WT	UTP	
5TFG	*H. sapiens*	1.91	WT	5-methyl-UTP	
5TFI	*H. sapiens*	1.89	WT	dGTP	
5TFJ	*H. sapiens*	1.85	WT	dCTP	
3SI7	*Mouse*	2.25	F508del	ATP	[28]
1XF9	*Mouse*	2.7	F508S	ATP	[7]
1XFA	*Mouse*	3.1	F508R	ATP	
1Q3H	*Mouse*	2.5	F508R	ANP	[29]
1R10	*Mouse*	3.0	WT *plus* R-domain	ATP	
1R0Z	*Mouse*	2.35	WT *plus* R-domain	ATP	
1R0Y	*Mouse*	2.55	WT *plus* R-domain	ADP	
1R0X	*Mouse*	2.2	WT *plus* R-domain	ATP	
1R0W	*Mouse*	2.2	WT apo form	-	

dTTP: thymidine-5′-triphosphate; dUTP: deoxyuridine-5′-triphosphate. *H. sapiens: Homo sapiens*

**Table 2 ijms-21-02407-t002:** Structural data for the complete wild type (WT) CFTR protein as determined by cryoelectron microscopy (cryo-EM) and listed in the PDB.

PDB Code	Organ.	Resol. (Å)	(Residues Count) and Features	in Complex with	Ref.
5UAK	*H. sapiens*	3.87	(1508), DP	-	[32]
6MSM	*H. sapiens*	3.2	(1506), P, 1 stabilizing mutation	ATP	[33]
6O1V	*H. sapiens*	3.2	(1489), DP, 1 solubilizing mutation	ATP and GLPG1837	[34]
6O2P	*H. sapiens*	3.2	(1489), DP, 1 solubilizing mutation	ATP and VX770	
6D3R	*G. gallus.*	4.3	(1437), DP, many stabilizing mutations	ATP	[35]
6D3S	*G gallus.*	6.6	(1437), P, many stabilizing mutations	ATP	
5UAR	*D. reiro.*	3.73	(1494), DP	-	[36]
5W81	*D. reiro.*	3.37	(1494), P, 1 stabilizing mutation	ATP	[37]

For all the structures reported, several regions of the protein (usually corresponding to highly flexible or disordered portions) remain undetermined because they can be hardly solved experimentally. DP: dephosphorylated; P: phosphorylated. *G. gallus: Gallus gallus. D. reiro: Danio reiro.*

**Table 3 ijms-21-02407-t003:** List of physiological binders evaluated by SPR for their capacity to interact with CFTR.

Natural Binder	CFTR Domain	K_d_ (μM)	Ref.
EBP50/NHERF1 PDZ1	C-ter (a.a. 1411–1480)	0.211–1.5 ^a^	[59]
PDZ2		0.267–4.8 ^a^	
PDZ1	C-ter (a.a. 1451–1480)	0.023	[60]
PDZ2		0.074	
PDZ1+2		0.022	
Shank2 PDZ	C-ter (a.a. 1451–1480)	0.056	[60]
CAP70 protein PDZ1	C-ter (EEVQDTRL)	0.22	[61]
PDZ2		0.008	
PDZ3		0.120	
AP-2	C-ter (KVIEENKVRQYDSIQ)	not determined	[62]
adaptor S100A10	WT NBD1L	7.8	[63]
NDPK-B	WT NBD1	not determined	[64]
Calumenin	WT full length	not determined	[65]
K8	WT NBD1	0.048	[66]
		0.04	
	F508del-NBD1	0.016	[67]
		0.02	[66]
K8 fragment (a.a. 83–105)	WT NBD1	31.0	[67]
	F508del-NBD1	4.6	[68]
cytokeratin 18	WT NBD1	no binding	[66]
	F508del-NBD1	no binding	
HSC70	WT NBD1	0.014	[14]
	F508del-NBD1	0.003	
ATP	NBD1	2.5	[23]
GroEL chaperonin	WT NBD1	0.025	[69]
annexin V	WT full length	not determined 0.002–0.004 ^b^	[70]
	WT NBD1		
Albumin	WT full length	no binding	[70]
human immunoglobulin G	WT NBD1 (a.a. 503–519)	0.069	[71]
	F508del-NBD1 (a.a. 503–518)	0.086	
	NBD2 (a.a.1237–1253)	no binding	

^a^ Data obtained with different subunit/peptide sequences of the indicated CFTR-binder. ^b^ The two Kd values were obtained in the presence or in the absence of calcium and ATP, respectively.

**Table 4 ijms-21-02407-t004:** List of natural or synthetic binders evaluated by SPR for their capacity to interact with CFTR.

Putative CFTR-Rescuing Molecules	CFTR Domain	K_d_ (μM)	Ref.
intact crotoxin	WT NBD1	0.004–0.118 ^a^	[72]
crotoxin subunit CBa_2_	F508del-NBD1	0.028	
*Viperidae* snake venom PLA_2_	WT NBD1	0.035	[72]
	F508del-NBD1	0.037	
Corr-4a	F508del-NBD1	not determined	[14]
VRT-325		not determined	
CFTRinh-172		no binding	
pyrazole compound 4172	F508del-NBD1	38.0	[73]
BIA		> 1,000	
aminothiazole compound 3152		no binding	
sulfamoyl-pyrrol compound 6258		no binding	
VX770	F508del-NBD1	no binding	[52]
Corr-4a		no binding	
VX809		24.2	
AAT compound 4		99.3	
AAT compound 5		40.3	
AAT compound 6		197.9	
AAT compound 7		no binding	
VX809	F508del-CFTR	72.8	[48]
Corr-4a		18.6	
VX661		206.1	
AAT compound 4		146.6	
AAT compound 5		19.7	
AAT compound 6		44.1	
AAT compound 7		4.5	
compound EN503		9.2	
anti-R-domain 1660 MAB	WT CFTR	not determined	[70]
anti-R-domain AG6 FAB	R-domain	0.032–0.009 ^b^	[74]
anti-NBD1 L12B4 MAB	WT NBD1	not determined	[14,70]

^a^ Data obtained with different subunit/peptide sequences of the indicated CFTR-binder. ^b^ The two Kd values were obtained for the unphosphorylated or phosphorylated form of the R-region, respectively. AAT: aminoarylthiazole; BIA: bromoindole-3-acetic acid; Corr-4a: corrector-4a; FAB: antigen-binding fragment; MAB: monoclonal antibody.

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
