# Peer review of "Recent Strategic Advances in CFTR Drug Discovery: An Overview"

_ijms, 2020, doi:10.3390/ijms21072407_

Round 1

Reviewer 1 Report

In the review the authors address the topics that are of great interest to scientists working on cystic fibrosis (CF). This manuscript is well-written with highly important up to date content. In my opinion, the manuscript is definitely worth to be published. The readers of IJMS should find the review worth reading. I have only one comment. Adding the note in the “5. Conclusions and perspectives” section about the perspectives of developing the amplifiers, which are able to increase the amount of CFTR in cells and tissues, would increase the value of the review.

Author Response

please see the cover letter attached

Reviewer 2 Report

The review investigates an interesting adn expanding topics. However, I suggest to the authors to modify the ms. Conclusion section is limited and does not offer much space to discuss latest advancments as well as possible real applications.

Author Response

please see the cover letter attached
